# Toward Adversarial Training on Contextualized Language Representation

**Hongqiu Wu**[1,2] **& Yongxiang Liu**[1,2] **& Hanwen Shi**[1,2] **& Hai Zhao**[1,2,*]**& Min Zhang**[3]

[1]Department of Computer Science and Engineering, Shanghai Jiao Tong University
[2]Key Laboratory of Shanghai Education Commission for Intelligent Interaction
and Cognitive Engineering, Shanghai Jiao Tong University, Shanghai, China
[3]School of Computer Science and Technology, Soochow University, Suzhou, China
{wuhongqiu,sam.liu,shihanwen}@sjtu.edu.cn,zhaohai@cs.sjtu.edu.cn,
minzhang@suda.edu.cn

## Abstract

Beyond the success story of adversarial training (AT) in the recent text domain on top of pre-trained language models (PLMs), our empirical study showcases the inconsistent gains from AT on some tasks, e.g. commonsense reasoning, named entity recognition. This paper investigates AT from the perspective of the contextualized language representation outputted by PLM encoders. We find the current AT attacks lean to generate sub-optimal adversarial examples that can fool the decoder part but have a minor effect on the encoder. However, we find it necessary to effectively deviate the latter one to allow AT to gain. Based on the observation, we propose simple yet effective *Contextualized representation-Adversarial Training* (CreAT), in which the attack is explicitly optimized to deviate the contextualized representation of the encoder. It allows a global optimization of adversarial examples that can fool the entire model. We also find CreAT gives rise to a better direction to optimize the adversarial examples, to let them less sensitive to hyperparameters. Compared to AT, CreAT produces consistent performance gains on a wider range of tasks and is proven to be more effective for language pretraining where only the encoder part is kept for downstream tasks. We achieve the new state-of-the-art performances on a series of challenging benchmarks, e.g. AdvGLUE ($59.1 \rightarrow 61.1$), HellaSWAG ($93.0 \rightarrow 94.9$), ANLI ($68.1 \rightarrow 69.3$).

## 1 Introduction

Adversarial training (AT) (Goodfellow et al., 2015) is designed to improve network robustness, in which the network is trained to withstand small but malicious perturbations while making correct predictions. In the text domain, recent studies (Zhu et al., 2020; Jiang et al., 2020) show that AT can be well-deployed on pre-trained language models (PLMs) (e.g. BERT (Devlin et al., 2019), RoBERTa (Liu et al., 2019)) and produce impressive performance gains on a number of natural language understanding (NLU) benchmarks (e.g. sentiment analysis, QA).

However, there remains a concern as to whether it is the adversarial examples that facilitate the model training. Some studies (Moyer et al., 2018; Aghajanyan et al., 2021) point out that a similar performance gain can be achieved when imposing random perturbations. To answer the question, we present comprehensive empirical results of AT on wider types of NLP tasks (e.g. reading comprehension, dialogue, commonsense reasoning, NER). It turns out the performances under AT are inconsistent across tasks. On some tasks, AT can appear mediocre or even harmful.

This paper investigates AT from the perspective of the *contextualized language representation*, obtained by the Transformer encoder (Vaswani et al., 2017). The background is that a PLM is typically composed of two parts, a Transformer-based encoder and a decoder. The decoder can vary from tasks (sometimes a linear classifier). Our study showcases that, the AT attack excels at fooling the

---
*Corresponding author; This paper was partially supported by Key Projects of National Natural Science Foundation of China (U1836222 and 61733011); https://github.com/gingasan/CreAT.

decoder, thus generating greater training risk, while may yield a minor impact on the encoder part. When this happens, AT can lead to poor results, or degenerate to random perturbation training (RPT) (Bishop, 1995).

Based on the observations, we are motivated that AT facilitates model training through robust representation learning. Those adversarial examples that effectively deviate the contextualized representation are the necessary driver for its success. To this end, we propose simple yet effective *Contextualized representation-Adversarial Training* (CreAT), to remedy the potential defect of AT. The CreAT attack is explicitly optimized to deviate the contextualized representation. The obtained "global" worst-case adversarial examples can fool the entire model.

CreAT contributes to a consistent fine-tuning improvement on a wider range of downstream tasks. We find that this new optimization direction of the attack causes more converged hyperparameters compared to AT. That means CreAT is less sensitive to the inner ascent steps and step sizes. Additionally, we always re-train the decoder from scratch and keep the PLM encoder weights for fine-tuning. As a direct result of that, CreAT is shown to be more effective for language pre-training. We apply CreAT to MLM-style pre-training and achieve the new state-of-the-art performances on a series of challenging benchmarks (e.g. AdvGLUE, HellaSWAG, ANLI).

## 2 PRELIMINARIES

This section starts by reviewing the background of adversarial training and contextualized language representation, and then experiments are made to investigate the impact of adversarial perturbations on BERT from two perspectives: (1) output predictions and (2) contextualized language representation. The visualization results lead us to a potential correlation between them and the adversarial training gain.

### 2.1 ADVERSARIAL TRAINING

Adversarial training (AT) (Goodfellow et al., 2015) improves model robustness by pulling close the perturbed model prediction and the target distribution (i.e. ground truth). We denote the output label as $y$ and model parameters as $\theta$, so that AT seeks to minimize the divergence (i.e. Kullback-Leibler divergence):

$$\min_{\theta} \mathcal{D}\left[q(y|\mathbf{x}), p(y|\mathbf{x} + \delta^*, \theta)\right] \tag{1}$$

where $q(y|\mathbf{x})$ refers to the target distribution while $p(y|\mathbf{x} + \delta^*, \theta)$ refers to the output distribution under a particular adversarial perturbation $\delta^*$. The adversarial perturbation is defined as the worst-case perturbation which can be evaluated by maximizing the empirical risk of training:

$$\delta^* = \arg\max_{\delta; \|\delta\|_F \leq \epsilon} \mathcal{D}\left[q(y|\mathbf{x}), p(y|\mathbf{x} + \delta, \theta)\right] \tag{2}$$

where $\| \delta \|_F \leq \epsilon$ refers to the decision boundary (the Frobenius norm) restricting the magnitude of the perturbation.

In the text domain, a conventional philosophy is to impose adversarial perturbations to word embeddings instead of discrete input text (Miyato et al., 2017). It brings down the adversary from the high-dimensional word space to low-dimensional representation to facilitate optimization. As opposed to the image domain, where AT can greatly impair model performances (Madry et al., 2018; Xie et al., 2019), such bounded embedding perturbations are proven to be positive for text models (Jiang et al., 2020; Zhu et al., 2020; Wang et al., 2021a). However, there is no conclusion as to where such a gain comes from.

### 2.2 CONTEXTUALIZED REPRESENTATION AND TRANSFORMER ENCODER

Transformer (Vaswani et al., 2017) has been broadly chosen as the fundamental encoder of PLMs like BERT (Devlin et al., 2019), RoBERTa (Liu et al., 2019), XL-Net (Yang et al., 2019), DeBERTa (He et al., 2021). The Transformer encoder is a stacked-up language understanding system with a number of self-attention and feed-forward sub-layers, which continuously place each token into the context and achieve its corresponding contextualized representation as its output.

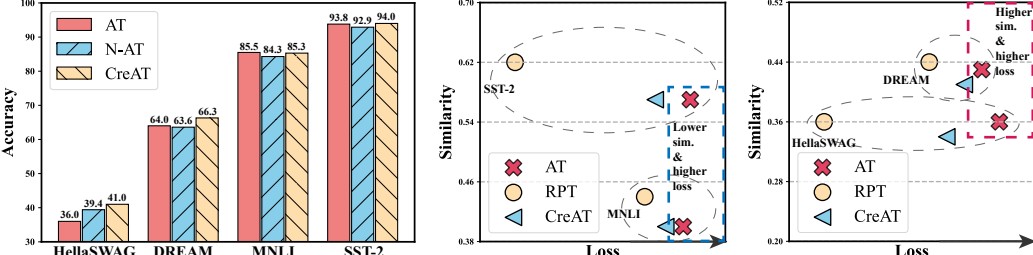

Figure 1: **Left:** Performances across different training methods, where N-AT refers to the regular situation where no adversarial training is applied. **Middle & right:** The relationship between the embedding similarity (contextualized representation) and training loss on different types of tasks. Note that we shift the training loss along the horizontal axis for a better view, the magnitude of which increases along the axis. We also investigate the attack success rate of AT in Appendix B.

One PLM will be deployed in two stages. For pre-training, the encoder will be trained on a large-scale corpus for a long period. For fine-tuning, the pre-trained encoder will be attached with a task-specific classifier as the decoder, which is initialized from scratch (we do not distinguish between the classifier and decoder in what follows). Eventually, the contextualized representation will be re-updated toward the downstream task.

## 2.3 IMPACT OF ADVERSARIAL TRAINING ON PLMS

To access the impact of AT, we choose four types of tasks and fine-tune BERT (Devlin et al., 2019) on them respectively: (1) sentiment analysis (SST-2), (2) natural language inference (MNLI), (3) dialogue comprehension (DREAM), (4) commonsense reasoning (HellaSWAG). Among them, the first two tasks are relatively simple, while the last two are much more challenging. We summarize the test accuracies across different training methods in Figure 1 (left).

We calculate two statistical indicators for each training process. The first is the training loss. A higher training loss indicates that the model prediction is more deviated from the ground truth after the perturbation. The second is the cosine similarity of the output embeddings before and after the perturbation. Researchers always take the final output of the encoder (i.e. output embeddings) as the representative of contextual language representation (Li et al., 2020a; Gao et al., 2021). Specifically, we consider the lower bound of all the tokens in the sentence. A lower similarity indicates that the encoder is more disturbed by the perturbation. Note that the indicators here are averaged over the first 20% of the training steps, as the model will become more robust under AT. We depict the above two indicators in Figure 1 (middle and right). The discussion is below. We also depict random perturbation training (RPT) and CreAT (will be presented later) as references here for better comparison with AT. We will mention this figure again in the next section.

• **Observation 1: The gain from AT is inconsistent on the four tasks.** From Figure 1 (left), we can see that AT achieves nice results on two sentence-level classification tasks (0.9 and 1.2 points absolute gain on MNLI and SST-2). On the other two tasks, AT performs badly, even leading to a 3.4 points drop on HellaSWAG.

• **Observation 2: AT contributes the most to deviating model predictions.** From Figure 1 (middle and right), we can see that the training loss caused by AT is always the highest (always on the rightmost side of the graph).

• **Observation 3: AT weakly influences the contextualized representation on DREAM and HellaSWAG.** We can see that the resultant similarities of the output embeddings caused by AT are also both high (equivalent to RPT on HellaSWAG and slightly lower than RPT on DREAM). It suggests that the adversarial examples are gentle for the encoder, although they are malicious enough to fool the classifier. However, we see an opposite situation on SST-2 and MNLI, where the similarities caused by AT are much lower.

The above observations raise a question on AT in the text domain: **whether fooling model predictions is purely correlated with training performances of PLMs?** Grouping observation 1 and observation 2, we see that high training loss does not always lead to better performances (i.e. accuracies). Grouping observation 2 and observation 3, we can see that AT mainly fools the classifier on some of the tasks, but takes little impact on the encoder.

Summing all observations up together, we can derive that **the performance gain from AT on PLMs is necessarily driven by the fact that the model needs to keep its contextualized representation robust from any adversarial perturbation so that its performance is enhanced**. However, this favour can be buried on tasks such as reading comprehension and reasoning, where AT is weakly-effective in deviating the contextualized representation (comparable to RPT). The resultant classifier becomes robust, while the encoder is barely enhanced.

## 3 CONTEXTUALIZED REPRESENTATION-ADVERSARIAL TRAINING

In this section, we propose *Contextualized representation-Adversarial Training* (CreAT) to effectively remedy the potential defect of AT.

### 3.1 DEVIATE CONTEXTUALIZED REPRESENTATION

We let $h(\mathbf{x}, \theta)$ be the contextualized representation (i.e. the final output of the encoder), where $\theta$ refers to the model parameters and $\mathbf{x}$ refers to the input embeddings. Similarly, we let $h(\mathbf{x} + \delta, \theta)$ be the contextualized representation after $\mathbf{x}$ is perturbed by $\delta$.

Our desired direction of the perturbation is to push away the two output states, i.e. decreasing their similarity. We leverage the cosine similarity to measure the angle of deviation between two word vectors (Mikolov et al., 2013; Gao et al., 2021). Thus, deviating the contextualized representation is the same as:

$$\min_{\delta} \mathcal{S}\left[h(\mathbf{x}, \theta), h(\mathbf{x} + \delta, \theta)\right] \tag{3}$$

where $\mathcal{S}[a, b] = \frac{a \cdot b}{\|a\| \cdot \|b\|}$.

### 3.2 CREAT

CreAT seeks to find the worst-case perturbation $\delta^*$ which deviates both the output distribution and contextualized representation, which can be formulated as:

$$\delta^* = \arg\max_{\delta; \|\delta\|_F \leq \epsilon} \mathcal{D}\left[q(y|\mathbf{x}), p(y|\mathbf{x} + \delta, \theta)\right] - \tau \mathcal{S}\left[h(\mathbf{x}, \theta), h(\mathbf{x} + \delta, \theta)\right] \tag{4}$$

where $\tau$ is the temperature coefficient to control the strength of the attacker on the contextual representation and a larger $\tau$ means that the attacker will focus more on fooling the encoder. CreAT is identical to AT when $\tau = 0$.

**Why CreAT?** In the previous discussion, AT is found to lead to the local worst case that partially fools the decoder part of the model. CreAT can be regarded as the "global" form of AT, which solves the global worst case for the entire model (both the encoder and decoder). The encoder is trained to be robust with its contextualized representation so that the model can perform better.

**Training with CreAT** Different from commonly-used AT methods (Goodfellow et al., 2015; Zhang et al., 2019b; Wang et al., 2020), where the adversarial risk is treated as a regularizer to enhance the alignment between robustness and generalization, in this paper, we adopt a more straightforward method to combine the benign risk and adversarial risk (Laine & Aila, 2017; Wang & Wang, 2022). Given a task with labels, the training objective is as follows:

$$\min_{\theta} \lambda \mathcal{L}(\mathbf{x}, y, \theta) + (1 - \lambda)\mathcal{L}(\mathbf{x} + \delta^*, y, \theta) \tag{5}$$

where $\mathcal{L}$ is the task-specific loss function and the two terms refer to the training loss under the respective benign example $\mathbf{x}$ and adversarial example $\mathbf{x} + \delta$. We find that $\lambda = 0.5$ performs just well

---

**Algorithm 1** Contextualized representation-Adversarial Training

---

**Input:** Model $\theta$, training set $T$, model step size $\beta$, ascent step size $\alpha$, decision boundary $\epsilon$, number of ascent steps $k$, temperature coefficient $\tau$

1: **while** not converged **do**
2:      $\{\mathbf{x}, y\} \leftarrow \text{SampleBatch}(T)$
3:      $\delta_0 \leftarrow \text{Init}()$
4:      $\text{Forward}(\mathbf{x}, \theta)$                                            $\triangleright$ Go benign forward pass
5:      **for** $j = 1$ to $k$ **do**
6:          $\text{Forward}(\mathbf{x} + \delta_{j-1}, \theta)$                       $\triangleright$ Go adversarial forward pass
7:          $\delta_j \leftarrow \text{BackwardUpdate}(\delta_{j-1}, y, \tau, \alpha, \epsilon)$     $\triangleright$ Update the perturbation following Eq. 4
8:      **end for**
9:      $\text{Forward}(\mathbf{x} + \delta^*, \theta)$                                        $\triangleright$ $\delta^* \leftarrow \delta_k$
10:     $\theta \leftarrow \text{BackwardUpdate}(\theta, y, \beta)$           $\triangleright$ Update the model parameters following Eq. 5
11: **end while**

---

Table 1: Results across different tasks over five runs. The average numbers are calculated based on the right side of the table (5 tasks). The variances for all tasks except WNUT and DREAM are low ($< 0.3$), so we omit them. Our CreAT-trained DeBERTa achieved the state-of-the-art result (94.9) on HellaSWAG (H-SWAG) on May 5, 2022[1].

| | MNLI-m (Acc) | QQP (F1) | WNUT (F1) | DREAM (Acc) | H-SWAG (Acc) | AlphaNLI (Acc) | RACE (Acc) | Avg | Gain |
|---|---|---|---|---|---|---|---|---|---|
| $\text{BERT}_{\text{base}}$ | 84.3 | 71.6 | $48.6_{0.8}$ | $63.0_{0.9}$ | 39.4 | 65.2 | 65.3 | 56.3 | - |
| + *FreeLB* | 85.5 | **73.1** | $48.2_{1.3}$ | $64.1_{0.9}$ | 39.8 | 65.3 | 62.8 | 56.4 | ↑ 0.1 |
| + *SMART* | **85.6** | 72.7 | $48.8_{0.8}$ | $64.5_{0.6}$ | 39.2 | 65.1 | 63.3 | 56.2 | ↓ 0.1 |
| + *AT* | 85.2 | 72.9 | $49.7_{1.2}$ | $63.8_{0.6}$ | 36.0 | 64.9 | 63.0 | 55.5 | ↓ 0.8 |
| + *CreAT* | 85.3 | 73.0 | $\mathbf{49.9}_{0.9}$ | $\mathbf{66.0}_{0.7}$ | **40.5** | **67.0** | **68.0** | **58.3** | ↑ **2.0** |

in our experiments. Eq. 5 is agnostic to both pre-training (e.g. masked language modeling $\mathcal{L}_{\text{mlm}}$) and fine-tuning (e.g. named entity recognition $\mathcal{L}_{\text{ner}}$) of PLMs.

Algorithm 1 summarizes the pseudocode of CreAT. The inner optimization is based on projected gradient descent (PGD) (Madry et al., 2018). At each training step, which corresponds to the outer loop (line 2 $\sim$ line 10), we fetch the training examples and initialize the perturbation $\delta_0$. In the following inner loop (line 5 $\sim$ line 7), we iterate to evaluate $\delta^*$ by taking multiple projected gradient steps. At the end of the inner loop, we obtain the adversarial perturbation $\delta^* = \delta_k$. Eventually, we train and optimize the model parameters with the adversarial examples (line 10).

## 4 EMPIRICAL RESULTS

Our empirical results include both general and robust-learning tasks. The implementation is based on *transformers* (Wolf et al., 2020).

### 4.1 SETUP

We conduct fine-tuning experiments on $\text{BERT}_{\text{base}}$ (Devlin et al., 2019). We impose the same bounded perturbation for all adversarial training methods (fix $\epsilon$ to 1e-1). We tune the ascent step size $\alpha$ from {1e-1, 1e-2, 1e-3} and the number of ascent steps $k$ from {1, 2} following the settings in previous papers (Jiang et al., 2020; Liu et al., 2020).

We conduct MLM-style continual pre-training and obtain two language models: $\text{BERT}_{\text{base}}^{CreAT}$ based on $\text{BERT}_{\text{base}}$ (Devlin et al., 2019), $\text{DeBERTa}_{\text{large}}^{CreAT}$ based on $\text{DeBERTa}_{\text{large}}$ (He et al., 2021). For the training corpus, we use a subset (nearly 100GB) of C4 (Raffel et al., 2020). Every single model is trained with a batch size of 512 for 100K steps. For adversarial training, we fix $\alpha$ and $\epsilon$ to 1e-1,

---

[1]https://leaderboard.allenai.org/hellaswag/submissions/public

Table 2: Results on AdvGLUE. For dev sets (upper), we report the results over five runs and report the mean and variance for each. For test sets (bottom), the results are taken from the official leaderboard, where CreAT achieved the new state-of-the-art on March 16, 2022[2].

| | SST-2 | QQP | QNLI | MNLI-m | MNLI-mm | RTE | Avg |
|---|---|---|---|---|---|---|---|
| | (Acc) | (Acc/F1) | (Acc) | (Acc) | (Acc) | (Acc) | |
| $\text{BERT}_{\text{base}}$ | $32.3_{1.4}$ | $50.8_{2.1}$/- | $40.1_{0.6}$ | $32.6_{1.3}$ | $19.3_{0.8}$ | $37.0_{2.5}$ | 35.4 |
| $\text{FreeLB-BERT}_{\text{base}}$ | $31.6_{0.3}$ | $51.0_{2.4}$/- | $\mathbf{45.4}_{0.3}$ | $33.5_{0.4}$ | $21.9_{0.9}$ | $42.0_{1.5}$ | 37.6 |
| $\text{BERT}_{\text{base}}^{\text{MLM}}$ | $32.0_{0.6}$ | $48.5_{1.3}$/- | $43.4_{1.4}$ | $27.6_{0.4}$ | $20.8_{0.5}$ | $\mathbf{45.9}_{2.5}$ | 36.4 |
| $\text{BERT}_{\text{base}}^{CreAT}$ | $\mathbf{35.3}_{1.0}$ | $\mathbf{51.5}_{1.2}$/- | $44.8_{0.6}$ | $\mathbf{36.0}_{0.8}$ | $\mathbf{22.0}_{0.8}$ | $45.2_{2.0}$ | $\mathbf{39.1}$ ↑2.7 |
| $\text{T5}_{\text{large}}$ | 60.6 | 63.0/**55.7** | 57.6 | 48.4 | 39.0 | 62.8 | 55.3 |
| $\text{SMART-RoBERTa}_{\text{large}}$ | 50.9 | 64.2/44.3 | 52.2 | 45.6 | 36.1 | 70.4 | 52.0 |
| $\text{DeBERTa}_{\text{large}}$ | 57.9 | 60.4/48.0 | 57.9 | **58.4** | **52.5** | **78.9** | 59.1 |
| $\text{DeBERTa}_{\text{large}}^{CreAT}$ | **63.5** | **67.5**/54.5 | **61.8** | 56.7 | 51.7 | 72.0 | **61.1** ↑2.0 |

and $k$ to 1. Training a base/large-size model takes about 30/100 hours on 16 V100 GPUs with FP16. Readers may refer to Appendix A for hyperparameters details.

To distinguish different training settings, for example, CreAT-$\text{BERT}_{\text{base}}$ means we fine-tune the original BERT model with CreAT; $\text{BERT}_{\text{base}}^{CreAT}$ means we pre-train the model and then regularly fine-tune it without adversarial training; $\text{BERT}_{\text{base}}^{\text{MLM}}$ means we regularly pre-train and fine-tune the model.

We describe a number of adversarial training counterparts below.

● **FreeLB** (Zhu et al., 2020) is a state-of-the-art fast adversarial training algorithm, which incorporates every intermediate example into the backward pass.

● **SMART** (Jiang et al., 2020) is another state-of-the-art adversarial training algorithm, which keeps the local smoothness of model outputs.

● **AT** (i.e. vanilla AT) refers to the special case of CreAT when $\tau = 0$. Both AT and CreAT are traditional PGD attackers (Madry et al., 2018).

## 4.2 RESULTS ON VARIOUS DOWNSTREAM TASKS

We experiment on MNLI (natural language inference), QQP (semantic similarity), two representative tasks in GLUE (Wang et al., 2019a); WNUT-17 (named entity recognition) (Aguilar et al., 2017); DREAM (dialogue comprehension) (Sun et al., 2019); HellaSWAG, AlphaNLI (commonsense reasoning) (Zellers et al., 2019; Bhagavatula et al., 2020); RACE (reading comprehension) (Lai et al., 2017).

Table 1 summarizes the results on $\text{BERT}_{\text{base}}$. First, each AT method works well on MNLI and QQP (two sentence classification tasks), and the improvement is quite similar. However, on the right side of the table, we can see AT appears harmful on certain tasks (average drop over BERT: 0.1 for SMART, 0.8 for AT), while CreAT consistently outweighs all the counterparts (absolute gain over BERT: 3.0 points on DREAM, 1.8 on AlphaNLI, 2.7 on RACE) and raises the average score by 2.0 points. Table 1 also verifies our previous results in Figure 1. From Figure 1, we can also see that the embedding similarities under CreAT are the lowest on all four tasks.

AT is sensitive to hyperparameters. For example, FreeLB obtains 36.6 and 39.8 on H-SWAG under different $\alpha$, 1e-1 and 1e-3. However, we find that CreAT is less sensitive to the ascent step size $\alpha$, and 1e-1 performs well on all datasets. In other words, the direction of the CreAT attack makes it easier to find the global worst case.

## 4.3 ADVERSARIAL GLUE

Adversarial GLUE (AdvGLUE) (Wang et al., 2021b) is a robustness evaluation benchmark derived from a number of GLUE (Wang et al., 2019a) sub-tasks, SST-2, QQP, MNLI, QNLI, and RTE.

---

[2]`https://adversarialglue.github.io/`

Different from GLUE, it incorporates a large number of adversarial samples in its dev and test sets, covering word-level, sentence-level, and human-crafted transformations.

Table 2 summarizes the results on all AdvGLUE sub-tasks. We see that CreAT-trained models consistently outperform fine-tuned ones by a large margin. In addition, we see that directly fine-tuning $\text{BERT}_{\text{base}}^{CreAT}$ works better than fine-tuning with FreeLB. To verify the effectiveness of CreAT-based adversarial pre-training, we pre-train $\text{BERT}_{\text{base}}^{MLM}$ based on MLM with the same number of steps (100K steps). From Table 2, we see that simply training longer leads to a limited robustness gain, while the CreAT-trained one is much more powerful (over $\text{BERT}_{\text{base}}^{MLM}$: 3.3 on SST-2, 6.2 on QQP, 8.4 on MNLI-m). Compared with other strong PLMs (Liu et al., 2019; Raffel et al., 2020), the further CreAT-based pre-training let $\text{DeBERTa}_{\text{large}}^{CreAT}$ achieves the new state-of-the-art result.

## 4.4 Adversarial SQuAD

AdvSQuAD (Jia & Liang, 2017) is a puzzling machine reading comprehension (MRC) dataset derived from SQuAD-1.0 (Rajpurkar et al., 2016), by inserting sentences into the paragraphs, replacing nouns and adjectives in the questions, generating fake answers similar to the original ones.

Table 3: Results on two AdvSQuAD dev sets over three runs.

| | AddSent (EM/F1) | AddOneSent (EM/F1) |
|---|---|---|
| $\text{BERT}_{\text{base}}$ | $59.3_{.4}/65.7_{.2}$ | $67.3_{.3}/74.3_{.4}$ |
| FreeLB-$\text{BERT}_{\text{base}}$ | $60.9_{.6}/67.4_{.3}$ ↑1.6/1.7 | $67.9_{.2}/74.8_{.6}$ ↑0.6/0.5 |
| $\text{BERT}_{\text{base}}^{CreAT}$ | $\mathbf{62.3}_{.3}/\mathbf{69.5}_{.2}$ ↑**3.0/3.8** | $\mathbf{69.5}_{.2}/\mathbf{76.5}_{.3}$ ↑**2.2/2.2** |
| $\text{DeBERTa}_{\text{large}}$ | $80.2_{.2}/86.5_{.2}$ | $83.3_{.5}/89.1_{.1}$ |
| $\text{DeBERTa}_{\text{large}}^{CreAT}$ | $\mathbf{81.6}_{.1}/\mathbf{87.3}_{.3}$ ↑**1.4/0.8** | $\mathbf{84.3}_{.3}/\mathbf{90.2}_{.2}$ ↑**1.0/1.1** |

From Table 3, we see CreAT leads to impressive performance gains by 1 to 3 points on all metrics.

## 4.5 Adversarial NLI

Adversarial Natural Language Inference (ANLI) (Nie et al., 2020) is an iteratively-strengthened robustness benchmark. In each round, human annotators are asked to craft harder samples to fool the model. In our experiment, each model is trained on the concatenated training samples of ANLI and MNLI.

Table 4: Test results of all rounds on ANLI over five runs.

| | Round 1 | Round 2 | Round 3 | Avg |
|---|---|---|---|---|
| $\text{BERT}_{\text{base}}$ | $55.1_{.4}$ | $44.5_{.6}$ | $42.5_{.7}$ | 47.4 |
| $\text{BERT}_{\text{large}}$ | $57.6_{.4}$ | $\mathbf{48.0}_{.5}$ | $43.2_{.9}$ | 49.6 |
| $\text{BERT}_{\text{base}}^{CreAT}$ | $\mathbf{59.2}_{.8}$ ↑**4.1** | $45.9_{.7}$ ↑**1.4** | $\mathbf{44.0}_{1.1}$ ↑**1.5** | $\mathbf{49.7}$ ↑**2.1** |
| $\text{DeBERTa}_{\text{large}}$ | $77.8_{.7}$ | $66.2_{.2}$ | $60.4_{.8}$ | 68.1 |
| $\text{DeBERTa}_{\text{large}}^{CreAT}$ | $\mathbf{78.7}_{.7}$ ↑**0.9** | $\mathbf{66.9}_{.1}$ ↑**0.7** | $\mathbf{62.3}_{.5}$ ↑**1.9** | $\mathbf{69.3}$ ↑**1.1** |
| $\text{InfoBERT}_{\text{large}}$ | 75.5 | 51.4 | 49.8 | 58.9 |
| $\text{ALUM}_{\text{large}}$ | 72.3 | 52.1 | 48.4 | 57.6 |

Table 4 summarizes the test results of all rounds. $\text{BERT}_{\text{base}}^{CreAT}$ is powerful and even surpasses $\text{BERT}_{\text{large}}$, outperforming by 1.6 points and 0.8 point on Round 1 and Round 3. Besides, DeBERTa significantly outperforms the previous state-of-the-art models InfoBERT (Wang et al., 2021a) and ALUM (Liu et al., 2020). However, $\text{DeBERTa}_{\text{large}}^{CreAT}$ further pushes the average score from 68.1 to 69.3, and especially on the most challenging Round 3, it leads to a 1.9 points gain on $\text{DeBERTa}_{\text{large}}$.

# 5 Analysis

## 5.1 Ablation Study

To access the gain from CreAT more clearly, in this part, we disentangle the CreAT perturbations level by level. To ensure the fairness of experiments, all the mentioned training methods follow the same optimization objectives in Eq. 4 and Eq. 5. Specifically, we first apply random perturbation training (RPT) on top of BERT fine-tuning. Then we apply AT, where the random perturbations become adversarial. $\text{CreAT}^{-}$ refers to the case where the attack is solely optimized to deviate the contextualized representation (i.e. removing the left term in Eq. 4).

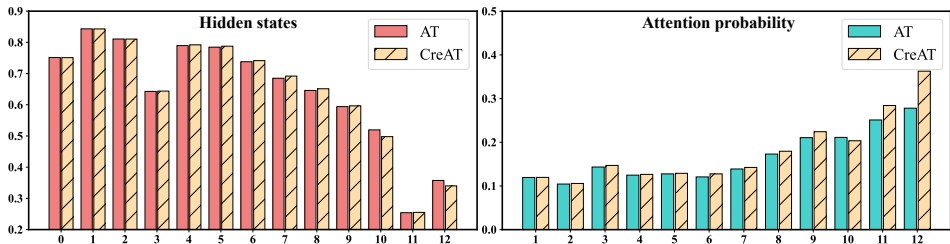

Figure 2: Impact of CreAT attack on the intermediate layers of BERT$_{\text{base}}$ (12 layers).

From Table 5, we can see that AT is superior to RPT on all three tasks. The improvement from CreAT is greater, outperforming AT by a large margin on DREAM and AlphaNLI. In addition, we can see CreAT$^-$ still improves DREAM and AlphaNLI without the adversarial objective. It turns out both parts of the CreAT attack are necessary. CreAT acts as a supplement to AT to find the optimal adversarial examples. The interesting thing is that removing the adversarial objective causes a large drop on MNLI. It implies that the gain from AT sometimes does not come from increasing the training risk, which can vary from tasks or datasets. We will leave this part for the future work.

Table 5: Results of ablation study over three runs.

|           | MNLI-m | DREAM | AlphaNLI |
|-----------|--------|-------|----------|
| BERT      | 84.3   | 63.0  | 65.2     |
| + *RPT*   | 84.1   | 63.6  | 64.9     |
| + *AT*    | 85.2   | 64.0  | 64.9     |
| + *CreAT$^-$* | 83.5 | 65.0 | 65.7    |
| + *CreAT* | **85.3** | **66.4** | **67.1** |

Table 6 demonstrates a number of adversarial pre-training methods. We can see that CreAT obtains stronger performances than simply training MLM for a longer period on all three tasks. However, the FreeLB-based one is almost comparable to MLM, while the AT-based one is slightly better than it. To explain, we only keep the encoder and drop the MLM decoder when fine-tuning the PLMs on downstream tasks, while CreAT can effectively allow a more robust encoder.

Table 6: Continual pre-training results on BERT$_{\text{base}}$ in different settings over fine runs. Each model is trained for the same number of steps.

|           | HellaSWAG | PAWS-QQP | PAWS-Wiki |
|-----------|-----------|----------|-----------|
| MLM       | 42.6      | 88.5     | 92.0      |
| + *FreeLB* | 42.7     | 88.2     | 92.1      |
| + *AT*    | 43.1      | 89.3     | 92.8      |
| + *CreAT* | **43.8**  | **90.2** | **93.1**  |

## 5.2 IMPACT OF CREAT ATTACK

We take a closer look into the learned contextualized representation of each intermediate BERT layer, including the hidden states and attention probabilities, to compare the impact of CreAT and AT. We calculate the cosine similarity of the hidden states and the KL divergence of the attention probabilities before and after perturbations.

From Figure 2 (left), CreAT and AT look similar on lower layers. For the last few layers (10 $\sim$ 12), CreAT causes a stronger deviation to the model (lower similarity). From Figure 2 (right), we observe that CreAT has a strong capability to confuse the attention maps. It turns out AT is not strong enough to deviation the learned attentions of the model. This helps explain why CreAT can fool the PLM encoder more effectively.

## 6 RELATED WORK

• **Language modeling:** Our work acts as the complement for AT on pre-trained language models (PLMs) (Vaswani et al., 2017; Devlin et al., 2019; Liu et al., 2019; Lan et al., 2020; Clark et al., 2020; Raffel et al., 2020; Brown et al., 2020; He et al., 2021). From the perspective of language pre-training, it is essentially a common improvement of the conventional pre-training paradigm, e.g. MLM (Devlin et al., 2019; Clark et al., 2020; Yang et al., 2019; Wu et al., 2022). From the

perspective of model architecture, it is parallel to strengthening encoder performances (Vaswani et al., 2017).

• **Adversarial attack:** Adversarial training (AT) (Goodfellow et al., 2015; Athalye et al., 2018; Miyato et al., 2019) is a common machine learning approach to create robust neural networks. The technique has been widely used in the image domain (Madry et al., 2018; Xie et al., 2019). In the text domain, the AT attack is different from the black-box text attack (Gao et al., 2018) where the users have no access to the model. Researchers typically substitute and reorganize the composition of the original sentences, while ensuring the semantic similarity, e.g. word substitution (Jin et al., 2020; Dong et al., 2021), scrambling (Ebrahimi et al., 2018; Zhang et al., 2019c), back translation (Iyyer et al., 2018; Belinkov & Bisk, 2018), language model generation (Li et al., 2020b; Garg & Ramakrishnan, 2020; Li et al., 2021). However, a convention philosophy of the AT attack is to impose bounded perturbations to word embeddings (Miyato et al., 2017; Wang et al., 2019b), which is recently proven to be well-deployed on PLMs and facilitate fine-tuning on downstream tasks, e.g. SMART (Jiang et al., 2020), FreeLB (Zhu et al., 2020), InfoBERT (Wang et al., 2021a), ASA (Wu & Zhao, 2022). ALUM (Liu et al., 2020) first demonstrates the promise of AT to language pre-training. Our work practices AT on more types of NLP tasks and re-investigates the source of gain brought by these AT algorithms.

• **Adversarial defense:** Our work act as a novel AT attack to obtain the global worst-case adversarial examples. It is agnostic to adversarial defense for the alignment between accuracy and robustness, e.g. TRADES (Zhang et al., 2019b), MART (Wang et al., 2020), SSL (Carmon et al., 2019).

• **Self-supervising:** AT is closely related to self-supervised learning (Madras et al., 2018; Hendrycks et al., 2019), since it corrupts the model inputs. On the other hand, it is parallel to weight perturbations (Wen et al., 2018; Khan et al., 2018; Wu et al., 2020), and contrastive learning (Hadsell et al., 2006) which relies on the corruption of model structures (Chen et al., 2020; Gao et al., 2021).

AT is still expensive at the moment, though there are works for acceleration (Shafahi et al., 2019; Zhang et al., 2019a; Zhu et al., 2020; Wong et al., 2020). This work is agnostic to these implementations. Another important line is to rationalize the behaviour of the attacker (Sato et al., 2018; Chen & Ji, 2022). In our work, we explain the impact of AT on contextualized language representation.

## 7 CONCLUSION

This paper investigates adversarial training (AT) on PLMs and proposes *Contextualized representation-Adversarial Training* (CreAT). This is motivated by the observation that the AT gain necessarily derives from deviating the contextualized language representation of PLM encoders. Comprehensive experiments demonstrate the effectiveness of CreAT on top of PLMs, which obtains the state-of-the-art performances on a series of challenging benchmarks. Our work is limited to Transformer-based PLMs and other architectures, vision models are not studied.

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

# A  TRAINING DETAILS

Table 7: Hyperparameters for pre-training.

|  | BERT$_{base}$ | DeBERTa$_{large}$ |
|---|---|---|
| Number of hidden layers | 12 | 24 |
| Hidden size | 768 | 1024 |
| Intermediate size | 3072 | 4096 |
| Number of attention heads | 12 | 16 |
| Dropout | 0.1 | 0.1 |
| Batch size | 512 | 512 |
| Learning rate | 5e-5 | 6e-6 |
| Weight Decay | 0.01 | 0.01 |
| Max sequence length | 256 | 256 |
| Warmup proportion | 0.06 | 0.06 |
| Max steps | 100K | 100K |
| Gradient clipping | 1.0 | 1.0 |
| Ascent step size | 1e-1 | 1e-1 |
| Decision boundary | 1e-1 | 1e-1 |
| Ascent steps | 2 | 2 |
| FP16 | Yes | Yes |
| Number of GPUs | 16 | 16 |
| Training period | 30 hours | 100 hours |

Table 8: Hyperparameters for fine-tuning BERT. For DeBERTa, we fix all hyperparameters the same, but set the learning rate to 1e-5 for all tasks. (dp: dropout rate, bsz: batch size, lr: learning rate, wd: weight decay, msl: max sequence length, wp: warmup, ep: epochs).

|  | (A)MNLI | QQP | WNUT | DREAM | H-SWAG | AlphaNLI | RACE | SQuAD |
|---|---|---|---|---|---|---|---|---|
| dp | 0.1 | 0.1 | 0.1 | 0.1 | 0.1 | 0.1 | 0.1 | 0.1 |
| bsz | 128 | 128 | 16 | 16 | 16 | 64 | 8 | 16 |
| lr | 3e-5 | 5e-5 | 5e-5 | 2e-5 | 2e-5 | 5e-5 | 2e-5 | 5e-5 |
| wd | 0.01 | 0.01 | 0 | 0 | 0 | 0 | 0 | 0 |
| msl | 128 | 128 | 64 | 128 | 128 | 128 | 384 | 384 |
| wp | 0.06 | 0.06 | 0.1 | 0.06 | 0.06 | 0.06 | 0.06 | 0.06 |
| ep | 3 | 3 | 5 | 6 | 3 | 2 | 2 | 2 |
| $\alpha$ | 1e-1 | 1e-1 | 1e-1 | 1e-1 | 1e-1 | 1e-1 | 1e-1 | 1e-1 |
| $\epsilon$ | 1e-1 | 1e-1 | 1e-1 | 1e-1 | 1e-1 | 1e-1 | 1e-1 | 1e-1 |
| $k$ | 1 | 1 | 2 | 1 | 1 | 1 | 1 | 1 |

