# OpenReview forum: "Toward Adversarial Training on Contextualized Language Representation"
_ICLR.cc/2023/Conference — ICLR 2023 poster_

### Official Review · Reviewer_Vm6d · 2022-10-14

**Confidence:** 2
**Correctness:** 3
**Technical Novelty And Significance:** 3
**Empirical Novelty And Significance:** 3
**Recommendation:** 8

**Clarity, Quality, Novelty And Reproducibility:**

This paper is not the clearest written, but I will try not to factor that into my review. I believe the approach is novel but I am not an expert in the adversarial robustness area.

**Strength And Weaknesses:**

Strengths:
* The analysis about how simple adversarial training doesn't work on all NLP tasks seems important for the community to know about. It seems like a good corollary to Vision results that say there is an explicit tradeoff between robustness and performance (Madry et al 2018), and NLP results showing that it is effective on some tasks (Jiang et al 2020, Zhu et al 2020, Wang et al 2021 as in the paper). The NLP results might be cherry-picked and perhaps misleading to readers.
* The approach seems to work well on a variety of different tasks.

Weaknesses:
* Conventional wisdom is that BERT was an undertrained model. I'm glad DeBERTa large was tried too, but I'm curious as to whether this same result would hold on e.g. RoBERTa Base.
* I'd like to see some analysis as to the cost of running CreAT / AT for a finetuning task (e.g. HellaSwag.) For instance, if CreAT is K times more expensive than conventional finetuning (I'm not sure I know the exact K) -- is it better to do adversarial training or ensemble together K models? Future work might make these approaches more efficient, but I think this analysis would be nonetheless important for practitioners.
* I'd be curious as to how well the model does if only the new contextualized-representation perturbation is factored in. A reasonable question would be whether this is what's really important for pretrained language models during finetuning.

**Summary Of The Paper:**

This paper studies adversarial training for pretrained NLP models. In adversarial training for NLP, perturbations are applied to the input word embeddings -- such that the prediction changes. The authors investigate this and find that while it sometimes helps, it hurts many tasks, most notably a 3.4 points drop on HellaSWAG.

Investigating this further, the authors propose a new method called CreAT, where the idea is that a perturbation must not only change the model's prediction, but also push away the contextualized representation from the unperturbed state. Experiments show that it enables BERT-like models to be finetuned better.

**Summary Of The Review:**

I'm slightly leaning towards accepting this paper, but I'd like to see a bit more contextualization of the cost of adversarial training, and how important the novel contextualized-representation-perturbation is. If those concerns are addressed I'd be happy to increase my score in revision.

---
Update post-author response: updating my score from 6->8.

---

### Official Review · Reviewer_bCxQ · 2022-10-23

**Confidence:** 4
**Correctness:** 2
**Technical Novelty And Significance:** 3
**Empirical Novelty And Significance:** 2
**Recommendation:** 5

**Clarity, Quality, Novelty And Reproducibility:**

Clarity: the clarity is good, and the paper is easy to understand.
Quality: the paper is somewhat flawed since the method is well-illustrated yet the experimental results are not convincing enough.
Novelty: the idea of this paper is clear yet is somewhat trivial considering that the representation-related AT term might not be the key factor influencing the experimental results.
Reproducibility: the paper should be easy to implement.


**Strength And Weaknesses:**

Strength:
A. the paper is clearly written and easy to understand

Weakness:
A. the experimental results are not convincing enough.
Generally, methods such as FreeLB and SMART, as well as a bunch of following adversarial training methods such as Token-Aware Virtual Adversarial Training (Li et al. 2021), adversarial training for large neural language models (Liu et al. 2020), Weighted Token-Level Virtual Adversarial Training in Text Classification (Sae-Lim et al. 2022) are exploring the effectiveness of adversarial training on general NLU tasks.
The proposed method seems only to obtain improvements on adversarial datasets as illustrated in Table 1.
The performances of methods like freeLB are lower than BERT-base results which are rather strange to me.
Considering that the hyper-parameter setting is actually very important in the adversarial training process, I am wondering if the hyper-parameters are well-searched in all the baseline methods and the proposed methods.

B. a hyper-parameter concern.
As mentioned, the hyper-parameter setup is important in the adversarial training process, which is also well-studied in Searching for an effective defender: benchmarking defense against adversarial word substitution (Li et al. 2021) where a freeLB++ method is proposed with the perturbation range much wider than the FreeLB method.
According to the FreeLB++ method, the perturbation set to 1e-1 compared with some figures like 3e-5 or 3e-4 in the original FreeLB setup, the defense performances are significantly improved.
As the proposed CreAT method uses 1e-1 for all their experiments, I wonder whether the robustness improvement is not influenced by the contextualized representation-related term in the AT process.


**Summary Of The Paper:**

This paper studies the adversarial training problem in the NLP field.
Specifically, they focus on the contextualized representations and propose an AT algorithm that calculates the perturbations based on the contextualized representations, not just the classification losses.
They first provide an analysis exploring the loss and the similarity to back their intuition to construct representation-based adversarial training.
They conduct experiments on both NLU benchmarks and robustness-concerned datasets and prove the effectiveness of their method.


**Summary Of The Review:**

As mentioned in the strength and weaknesses, the paper is clear yet the experimental results seem less convincing especially since the key hyper-parameters are all fixed in all experiments.
I suggest a more detailed hyper-parameter search and analysis which could significantly improve the soundness of the proposed method.

---

### Official Review · Reviewer_ANTZ · 2022-10-25

**Confidence:** 2
**Correctness:** 4
**Technical Novelty And Significance:** 3
**Empirical Novelty And Significance:** 3
**Recommendation:** 6

**Clarity, Quality, Novelty And Reproducibility:**

Clarity:
The paper is well-written and easy to follow.

Novelty:
The work brings some new ideas on how to train a robust model.

Reproducibility:
Since the datasets are public and the authors describe their algorithm clearly, it should be easy for others to implement the experiments in the work.

**Strength And Weaknesses:**

Strength:
a new adversarial training approach that is effective over a number of tasks.




**Summary Of The Paper:**

The paper proposes a new adversarial training approach in which perturbation is added to the inner layers of a deep transformer and thus affects contextualized representations. Empirical studies are conducted over a number of benchmarks with remarkable improvements to the existing AT methods.

**Summary Of The Review:**

The paper discloses that existing AT methods fail to manipulate the contextual representations, and thus render inconsistent performance over different tasks. Hence, the authors consider add perturbations to inner layers of a deep transformer.

The observations are interesting, and the proposed method is simple yet effective. The descent improvements over a variety of tasks indicate that it is worth to try the method in practice. It would be better if the authors could release their code and thus other can reproduce the results more easily.

---------------------------------------
After reading the comments from other reviewers and the response from the authors, I agree that the improvements on BERT over the baselines are sometimes limited (e.g., on MNLI and QQP). I still feel that the idea is interesting, but also raise concerns on when or on what kind of tasks and models, the proposed method is really useful. Therefore, I slightly downgrade my rating.

---

### Decision · Program_Chairs · 2023-01-20

**Decision:**

Accept: poster

**Justification For Why Not Higher Score:**

The work would benefit from clearer and better-tuned baselines when ablating with FreeLB (particularly the epsilon parameter). In the new results, the pre-training and fine-tuning procedures are unclear.

**Justification For Why Not Lower Score:**

The work has a useful contribution in showing why adversarial training doesn't work for some NLP tasks. The proposed approach itself also seems to effective on a wide variety of tasks, is well ablated and also does well against an ensemble of the same cost.

**Metareview: Summary, Strengths And Weaknesses:**

This work proposes a new adversarial training approach where perturbations are added to the inner layers of the Transformer to affect an input's contextualized representation. Empirical experiments are run on NLU and robustness-based datasets that show the approach is effective relative to standard adversarial training.

Strengths:
* The paper analyses why simple adversarial training doesn't work for all NLP tasks.
* The proposed approach performs well on a variety of tasks.
* The paper is clearly written and easy to understand.
* The approach is well ablated in terms of cost and compared to an ensemble of the same cost.

Weaknesses:
* The improvements on BERT over the baselines (FreeLB) are partly limited (in particular MNLI and QQP). This raises concerns about which tasks the model really works for.
* The baselines (in particular FreeLB's epsilon parameter) might be undertuned leading to an unfair comparison.
* The work would benefit from a clear ablation between CreAT during pre-training and the FreeLB-style during pretraining. It's unclear how the experiment is constructed in the new result, i.e. what pretraining and finetuning procedures were used.
* The work would also benefit from showing the parameter search during finetuning.

**Note From Pc:**

if the above contains the word "oral" or "spotlight" please see: "oral" presentation means -> notable-top-5% and "spotlight" means -> notable-top-25%. As stated in our emails, we are disassociating presentation type from AC recommendations